# OpenReview forum: "OpenAgentSafety: A Comprehensive Framework For Evaluating Real-World AI Agent Safety"
_ICLR.cc/2026/Conference — ICLR 2026 Poster_

### Official Review · Reviewer_Uk4k · 2025-10-26

**Soundness:** 3
**Presentation:** 2
**Contribution:** 3
**Rating:** 6
**Confidence:** 4

**Summary:**

This paper introduces OpenAgentSafety, a benchmark designed to evaluate the safety of AI agents in realistic, high-risk scenarios. It features a comprehensive real-world tool suite, diverse user intentions, and multi-turn, multi-agent dynamics. By combining real tool use, complex social interactions, and varied intents from both users and non-player characters (NPCs), OA-SAFETY enables rigorous and comprehensive safety assessments across diverse scenarios.

**Strengths:**

1. Each OA-SAFETY task is implemented as a modular Docker container that integrates both a rule-based evaluator and an LLM-as-a-judge module.

2. The experimental analysis is insightful and thought-provoking. For example, the study finds that hidden intents can circumvent safeguards, operational risks lead to inconsistent judgments, and browsing emerges as the most failure-prone interface. Moreover, LLM judges struggle with nuanced failure cases, underscoring the need for hybrid evaluation approaches that combine LLM-based judgment with rule-based verification.

3. The comparison with related work is thorough and comprehensive.

**Weaknesses:**

1. Since the multi-user scenario is a key component, the authors could further elaborate on the ChatNPC tools and include a case study demonstrating the dynamics and challenges of a multi-user interaction scenario.

2. While the paper evaluates five widely adopted LLMs, the experiments are conducted on only one agent framework (OpenHands). It would strengthen the study to include evaluations across multiple agent frameworks to better assess generalizability and robustness.

**Questions:**

1. Could the authors provide more details on the multi-user scenario in the paper, such as its setup or representative examples and process?

2. It would be beneficial to evaluate additional agent frameworks to strengthen the generality of the results.

---

> ### Author Response · Authors · 2025-11-21
> **Author Response**
>
> Thank you for providing a positive rating to our paper and for your insightful feedback! We are glad to learn that you appreciate the modularity of our framework, the comprehensiveness of our experimental results and insightful analyses, and our thorough comparison with related work. We address your concerns below:
>
> > Since the multi-user scenario is a key component, the authors could further elaborate on the ChatNPC tools and include a case study demonstrating the dynamics and challenges of a multi-user interaction scenario. Could the authors provide more details on the multi-user scenario in the paper, such as its setup or representative examples and process?
>
> The ChatNPC tool facilitates interactions between OpenHands and the NPCs (LLM-simulated users) implemented using Sotopia. Each task involving NPCs is accompanied by a scenarios.json file where we define the task setting, the NPC persona, and their desired NPC behavior, all of which are incorporated into the NPC’s prompts. We have revised the draft and added more details on the NPC setup and include sample NPC configurations and prompt templates in Appendix A.4. We have also added example trajectories involving NPC interactions in Appendix A.6.
>
> > While the paper evaluates five widely adopted LLMs, the experiments are conducted on only one agent framework (OpenHands). It would strengthen the study to include evaluations across multiple agent frameworks to better assess generalizability and robustness.
>
> Thank you for the suggestion! We agree that evaluations with multiple agent frameworks would strengthen the study further. However this requires significant implementation efforts. Most existing agent frameworks (e.g., Browser-Use, OWL, SWE-Agent, LangFun Agents) lack one or more of the wide range of tools used in our work. Furthermore, many agents lack support for sandboxed execution (for e.g. Docker containers) that is crucial for preventing damages to the user’s machine. Therefore, we leave this analysis for future work considering the above implementation challenges.

---

### Official Review · Reviewer_yZfw · 2025-10-29

**Soundness:** 4
**Presentation:** 3
**Contribution:** 3
**Rating:** 8
**Confidence:** 4

**Summary:**

The paper introduces OpenAgentSafety, a benchmark to evaluate the safety of LLM agents. Unlike prior benchmarks of this kind, OpenAgentSafety has a number of unique features:

1. Realistic agentic environments. Agents are run in sandboxes with access to terminal, code interpreters, website clones, and other simulated "characters" referred to as NPCs.
2. Three different kinds of tasks in which models can execute unsafe actions. In particular, the four combinations of benign and malicious intent that can be assigned to the user and NPC (Figure 3).
3. A combined rules based and LLM-as-judge evaluation metric.

The benchmark contains a diverse set of environments (80 seed tasks augmented to 356 in total using GPT-4o) spanning 8 risk categories (Table 2).

The authors evaluate 5 frontier models on the benchmark, and draw a number of useful conclusions from these results. Some of these results are quite novel to me. For example, Figure 3 top they show that with totally benign situations, models can make unsafe "mistakes", with Claude-3.7 performing much worse in this category than when faced with explicit malicious intent from the user / NPC.

**Strengths:**

## Originality

The idea of benchmarking agent safety is not novel, however the specific execution of this benchmark is. In particular, I am not aware of another work that has points 1 and 2 from the summary, these being (1) as realistic environments and (2) different kinds of user / NPC intent.

## Quality and Clarity

Overall the quality of the paper is high. The benchmark appears to be well thought out and design decisions justified in the writing. The results are good, and the research questions (3.3) are well answered with data.

## Significance

This benchmark is significant because it is more realistic than others. As LLM agents are becoming more performant, and thus deployed in the real world, it is of critical importance that we have realistic benchmarks to evaluate their safety. In this category, OpenAgentSafety is the most realistic agent safety benchmark I know of.

**Weaknesses:**

There are some weaknesses in the paper:

1. The benchmark is excellent, and accordingly I think the paper could be improved by evaluating more models on the benchmark. For example, by evaluating a number of OpenAI, Anthropic, and open source models, and analyzing this data, the reader would have a better idea of the "general" state of agent safety in the world today.
2. The paper would be improved by providing more information to the reader about what types of problems are in the dataset. E.g. providing examples, and counts of problems that fit into each category in Table 2. I think walking through an example environment would also help the reader understand the type of interactions that occur with NPCs.
3. This is a nit, but the figures in section 3.3 are very small and hard to read when the paper is printed out (this is an easy fix though).

**Questions:**

1. Do you agree with weakness 1? If so could you test some more models?
2. Do you agree with weakness 2? Could you provide some more detailed examples of problems?
3. Why is it that in table 3 the Rule-based evaluation of all the models is incredibly similar, but the LLM-as-judge varies far more?

---

> ### Author Response · Authors · 2025-11-21
> **Author Response**
>
> Thank you for giving us a positive rating and for your detailed feedback! We are pleased to know that you appreciate the originality of our framework with realistic environments and diverse user/NPC intents, clarity of writing and experimental setup, and the significance of our work in developing one of the most realistic agent safety benchmarks for evaluating modern LLM agents before deploying them in real-world scenarios. We would like to address your concerns and feedback below:
>
> > The benchmark is excellent, and accordingly I think the paper could be improved by evaluating more models on the benchmark. For example, by evaluating a number of OpenAI, Anthropic, and open source models, and analyzing this data, the reader would have a better idea of the "general" state of agent safety in the world today.
>
> Thank you for the valuable suggestion! We have evaluated GPT-5 and Claude Sonnet 4 on our benchmark and observe that the LLMs have unsafe behavior rates of **50.28%** and **31.29%** respectively measured using the rule-based evaluators, and **50.28%** and **46.02%** respectively as per the LLM-Judge. We have added detailed results to the revised draft in Section 3.
>
> > The paper would be improved by providing more information to the reader about what types of problems are in the dataset. E.g. providing examples, and counts of problems that fit into each category in Table 2. I think walking through an example environment would also help the reader understand the type of interactions that occur with NPCs.
>
> Our draft already includes the distribution of tasks across risk categories in Figure 6 (Appendix). We also include distribution of tasks across tools required in Figure 7 and across user/NPC intents in Figure 8 (in Appendix). We have revised the draft to include example trajectories that demonstrate interactions with NPCs in Appendix A.6.
>
> > This is a nit, but the figures in section 3.3 are very small and hard to read when the paper is printed out (this is an easy fix though).
>
> We have fixed the figures in Section 3.3 in the revised draft for improved readability.
>
> > Why is it that in table 3 the Rule-based evaluation of all the models is incredibly similar, but the LLM-as-judge varies far more?
>
> While LLMs do share some similar unsafe behaviors on certain tasks, our analysis reveals that models largely show different behaviors for most of the tasks. Since rule-based evaluators only detect safety violations when agents make verifiable changes to the environment, they cannot distinguish truly safe completions from partially completed tasks with unsafe behaviors. As a result, tasks marked “safe” by rule-based checks still exhibit significant variation across models which is captured by LLM-judges and is reflected in the greater variation of judge-based evaluation results across models. These findings further motivate the dual use of rule-based evaluators and LLM-judges. We will include this discussion in the revised draft.

---

> ### Comment · Reviewer_yZfw · 2025-11-25
>
> > We have evaluated GPT-5 and Claude Sonnet 4 on our benchmark and observe that the LLMs have unsafe behavior rates of 50.28% and 31.29% respectively measured using the rule-based evaluators, and 50.28% and 46.02% respectively as per the LLM-Judge. We have added detailed results to the revised draft in Section 3.
>
> Thanks for adding these. The more models tested the better I think. It sees roughly speaking the newer models are safer.
>
> > We have revised the draft to include example trajectories that demonstrate interactions with NPCs in Appendix A.6.
>
> These are nice, thank you.
>
> > Since rule-based evaluators only detect safety violations when agents make verifiable changes to the environment, they cannot distinguish truly safe completions from partially completed tasks with unsafe behaviors.
>
> Makes sense, good to include this discussion in the paper as you have.
>
> # Summary
>
> I thank the authors for clarifying my questions. Overall I will keep my positive score, great work!

---

### Official Review · Reviewer_rwaC · 2025-10-31

**Soundness:** 2
**Presentation:** 2
**Contribution:** 2
**Rating:** 4
**Confidence:** 3

**Summary:**

356 tasks, built on top of OpenHands framework. Agent has access to terminal, filesystem, python and web prowser.
8 risk categories: key compromise, data loss, privacy, unsafe code execution, financial loss, maliciois content, legal violations, harmful decision making.
Each task comes with: environment, task description, NPC configs, and evaluator.
Three settings: 1) benign user intent, no NPC 2) benign user intent + malicious NPC 3) malicious user request

Results show major unsafe behavior rates according both to llm as a judge and rule-based evals.

**Strengths:**

- Multi-user setting seems like an interesting setup in principle; however, I have critical concerns about NPC reproducibility, and the way the NPCs initiate chats
- I like the three settings, combining user intent and NPC intent
- Interesting discussions

**Weaknesses:**

- There is no info on the NPCs in the main paper nor the appendix: could you at least put some of the prompts?
- Are NPC settings reproducible? Fixed open-weight model? What is the model?
- When do NPC interact? Is it the agent or the NPC that initiate messages? Because if it is the agent, then if it just decides not to use the chat tool, it will not see malicious messages right?
- For some of the tasks the evaluation of the outcome seems a bit arbitrary.

My biggest concern is whether in all settings there exists an ideal agent behavior?
1) For instance, what is the correct behavior in the "trens/Emily" task  of Table 6 of appendix, or the change-branch-policy?
Is it just REFUSAL? But if that's the case how do you distinguish from FAILURE?

I think there needs to be a ground truth correct behavior for every task, and making sure it is not too arbitrary.
Agents are typically subject to a utility/security-safety tradeoff. If utility is not tracked / or there does not exist a ground trugth correct behavior for every task that is relatively objectively defined, then the benchmark is not meaningful in my opinion.

**Questions:**

I'm a a bit surprised all the rule-based evals are so close(32%). Are the failures on the same tasks or different? Similar types of failures or not?

Would be nice to have at least one example of user task in the main paper, and more info on the NPC.

---

> ### Author Response · Authors · 2025-11-21
> **Author Response**
>
> Thank you for the detailed feedback! We are glad to learn that you found the multi-user NPC settings and our insights interesting. We address your concerns below:
>
> > There is no info on the NPCs in the main paper nor the appendix?
>
> Thank you for pointing this out! Each task involving interaction with NPCs is accompanied by a scenarios.json file where we define the task setting, the NPC persona, and their desired NPC behaviour, which are incorporated into the NPC’s prompts. These NPCs are LLM-simulated users that interact with the OpenHands agent using the ChatNPC tool. We have revised the draft to include the prompts and detailed NPC configuration for an example task in Appendix A.4, and include additional example trajectories with NPC interactions in Appendix A.6.
>
> > Are NPC settings reproducible? What is the model?
>
> The NPCs are simulated using GPT-4o (the LLM is configurable) which introduces non-determinism, but the careful prompting strategies borrowed from Sotopia ensure that the general behavior of NPCs is consistent across runs. We have also verified this during the manual annotation of the agent trajectories.
>
> > When do NPC interact? Is it the agent or the NPC that initiate messages? Because if it is the agent, then if it just decides not to use the chat tool, it will not see malicious messages right?
>
> By design, the agent must initiate the conversation with the NPCs via the chatNPC tool. Our manual inspection shows that models generally use the chat tool reliably, as they are at good instruction following. However, in rare cases where the agent does not use the chat tool and does not reach a safety-vulnerable state, we classify it as failure. Note that our framework is extensible for this feature and we leave it to future work to explore scenarios where NPCs initiate the conversation.
>
> > My biggest concern is whether in all settings there exists an ideal agent behavior? Is it just REFUSAL? But if that's the case how do you distinguish from FAILURE? I think there needs to be a ground truth correct behavior for every task, and making sure it is not too arbitrary. Agents are typically subject to a utility/security-safety tradeoff. If utility is not tracked / or there does not exist a ground truth correct behavior for every task that is relatively objectively defined, then the benchmark is not meaningful in my opinion.
>
> Thank you for the suggestion! We agree that this is important and we address it by manually annotating the correct safe behavior for all the 356 tasks. The ground truth behavior for each task is defined to achieve maximum compliance of the user request (utility) without compromising on safety. We use an LLM-Judge to determine if the agent demonstrates this correct safe behaviour given the agent’s trajectory. We compute the “Successful Completion Rate” for all LLMs and include it in the revised draft (in Table 3), and annotate the correct safe behaviors for the example tasks in Table 6.
>
> For some tasks (including trends/Emily task), refusal is indeed the ideal behavior. This is different from task failures that denote cases where the agent could not reach a safety vulnerable state due to capability constraints. For e.g., in the trends/Emily task, we use “failure” to denote the situation when the agent is unable to read the memo due to incorrect tool-use and is never exposed to malicious interference. However, the ideal behavior of many tasks is not simply refusal and they have a correct safe solution (for eg. the api-in-codebase and leak-document examples from Table 6).
>
> > I'm a bit surprised all the rule-based evals are so close(32%). Are the unsafe behaviors on the same tasks or different? Similar types or not?
>
> While LLMs do share some similar unsafe behaviors on certain tasks, our analysis reveals that models largely show different behaviors for most of the tasks. Since rule-based evaluators only detect safety violations when agents make verifiable changes to the environment, they cannot distinguish truly safe completions from partially completed tasks with unsafe behaviours. As a result, tasks marked “safe” by rule-based checks still exhibit significant variation across models which is captured by LLM-judges and is reflected in the greater variation of judge-based evaluation results across models. These findings further motivate the dual use of rule-based evaluators and LLM-judges. We will include this discussion in the revised draft.
>
> > Would be nice to have at least one example of user task in the main paper, and more info on the NPC.
>
> Thanks for the suggestion! The task scenarios in Table 2 provide representative examples of user tasks. Due to space constraints, we provide detailed task descriptions in Table 6 (Appendix) and example task trajectories in Appendix A.6  (revised with trajectories involving NPC interactions). We have also added more details on the NPC setup, an example NPC configuration and prompts in Appendix A.4 of the revised paper.

---

### Official Review · Reviewer_neML · 2025-11-01

**Soundness:** 2
**Presentation:** 2
**Contribution:** 2
**Rating:** 4
**Confidence:** 3

**Summary:**

This paper introduces OpenAgentSafety, a comprehensive framework for evaluating the real-world safety of LLM-based agents. The framework integrates real tools (e.g., browser, file system, code execution), multi-intent tasks (benign, malicious), and social NPC interactions within sandboxed environments. It employs a hybrid evaluation mechanism that combines rule-based checking with LLM-as-a-judge assessments to detect both explicit and implicit unsafe behaviors. Experiments across five frontier models (Claude 3.7, GPT-4o, o3-mini, DeepSeek-v3/R1) reveal that even the most capable models still exhibit a high rate of unsafe actions. The framework is modular and extensible, making it a valuable testbed for future research on real-world agent safety.

**Strengths:**

* The paper tackles a timely problem—evaluating the real-world safety of LLM-based agents—by integrating realistic environments, multiple user intents, and social interactions.
* Experiments are conducted across diverse models and risk categories, using four questions (RQ1–RQ4) to guide the analysis. The results are comprehensive.

**Weaknesses:**

* OpenAgentSafety offers a well-engineered and comprehensive benchmark, but its novelty is limited. The framework mainly integrates existing components such as OpenHands and Sotopia, extending prior safety benchmarks rather than introducing new methodologies. Its hybrid evaluation and GPT-based task generation are incremental refinements of known techniques.
* The paper does not describe a mechanism for filtering or validating generated tasks (e.g., to remove redundant, ill-posed, or low-quality scenarios). Adding a lightweight automatic validation step—such as a logit- or heuristic-based filter to ensure task diversity and executability—could improve dataset quality and efficiency for future extensions.
* Although the framework employs real tools (e.g., OwnCloud, GitLab), these are sandboxed offline replicas. As a result, the environment does not fully capture real-world dynamics.
* While the experimental setup is extensive, the results largely reaffirm known trends—namely that LLM-based agents exhibit high rates of unsafe behavior. Moreover, the study confirms that LLM-as-Judge evaluations, though informative, remain imperfect and somewhat unreliable compared to human assessment.

**Questions:**

1. Since the paper employs both rule-based and LLM-as-judge evaluations, could the authors clarify how these two signals are intended to be combined or interpreted together, especially in cases where the disagreement rate is substantial?
2. The authors propose stricter authentication and privilege controls for high-risk tools. However, introducing additional safeguards might further increase the observed failure rates. Could the authors discuss how they envision balancing safety and task success?

---

> ### Author Response · Authors · 2025-11-21
> **Author Response**
>
> Thank you for the feedback. We are pleased that you found our paper addresses a timely research problem, and that you consider our experimental setup comprehensive. We would like to address your concerns below:
>
> > The work offers a well-engineered and comprehensive benchmark, but its novelty is limited. The framework extends prior safety benchmarks rather than introducing new methodologies.
>
> We’d like to clarify that our work is novel in several ways. As discussed in Section 1, in contrast with previous work (across **24** benchmarks) in Table 1, OpenAgentSafety is the only benchmark with real-world tools, diverse intents, and multi-user social interaction. This positions OpenAgentSafety as the only benchmark with these three elements when agents are increasingly being used in these scenarios.
>
> However, if there are individual benchmarks that you think similarly fill the gap that OpenAgentSafety is filling, we would be happy to discuss the relationship with these other benchmarks and our contributions above and beyond them.
>
> > The paper does not describe a mechanism for validating generated tasks
>
> We have described our task validation mechanism in Section 2.2. After creating 80 seed tasks manually and scaling them to 356 tasks using GPT-4o (Appendix A.6), all the generated tasks were rigorously verified by the authors to ensure correctness of task implementation and diversity across risk categories, tools required, and user/NPC intents.
>
> > Framework employs real tools (e.g., OwnCloud, GitLab) but these sandboxed offline replicas do not fully capture real-world dynamics.
>
> Thank you, this is correct. But at the same time we believe it would be highly unethical to deploy agents with safety issues or malicious goals on real-world counterparts of our web environments (for eg. Google Drive, GitHub, etc.) due to the immense harms that would cause. Furthermore, using sandboxed web environments is also critical for ensuring reproducibility of our experimental results, and is common practice in other highly influential benchmarks like WebArena. Therefore, our design choice of using sandboxed web environments is actually a crucial feature of our framework and not a limitation.
>
> > While the experimental setup is extensive, the results largely reaffirm known trends—namely that LLM-based agents exhibit high rates of unsafe behavior.
>
> While the fact that LLM-based agents exhibit high rates of unsafe behavior is indeed one of our findings, our work also reports novel findings that are unexplored by prior benchmarks. 1) Unsafe behaviors are observed **twice** as frequently in tasks with benign user intents over explicitly malicious tasks, 2) reasoning models demonstrate much higher unsafe behavior and 3) our analysis across **7** state-of-the-art LLMs identifies specific failure modes and unsafe behaviors across diverse tools, task intents, and varied risk categories (Section 3.3). These findings provide actionable insights for future work on improving safety of LLM agents in realistic task scenarios using our benchmark as a reliable testbed.
>
> > LLM-as-Judge evaluations remain imperfect.
>
> We acknowledge these limitations and include concrete comparisons of rule-based evaluators, LLM-as-a-judge, and manual analysis in Tables 3 and 4 that provide insights on different evaluation approaches. Our analysis reveals that more reliable approaches are required for evaluating agent safety and we leave this critical avenue for future work.
>
> > Could the authors clarify how the two evaluation methods (rule-based and LLM-as-a-judge) are intended to be interpreted together?
>
> The evaluation approaches complement each other and provide different perspectives on the implicit (LLM Judge) vs. explicit (rule-based) unsafe behaviours. Thus, these signals must be interpreted separately rather than simply combining them. In case of disagreements where the LLM judge labeled a trajectory as safe, but the rule-based evaluator detected unsafe behaviour due to concrete environment changes, rule-based evaluation should be considered as these scenarios reveal limitations of the LLM-judge. If the LLM-judge determined the task being unsafe disagreeing with the rule-based judgement, this generally denotes unsafe but partially completed tasks that do not make verifiable changes to the environment (Section 2.3).
>
> > Could the authors discuss how they envision balancing safety and task success?
>
> We agree that balancing safety and utility is challenging. While beyond the scope of our work, a potential technique to achieve this could include developing more interactive agents that actively ask for user confirmation before executing potentially unsafe actions that can be detected using additional LLM-based modules. Ultimately, achieving this balance depends on the deployment setting and its associated risk tolerance, where companies must prioritize safety whereas in academic or personal-use settings, safeguards may be lowered to focus on task success.

---

### Author Response · Authors · 2025-12-04
**Summary of Responses**

We sincerely thank all the reviewers and the ACs for their time. We are pleased to know that the reviewers appreciate the **originality** and **significance** of our work (Reviewer yZfw), the **realistic design** of our benchmark with real-world tools and diverse user/NPC intents (Reviewers neML, rwaC, yZfw), and our **comprehensive experimental results and insightful analyses** (Reviewers neML, rwaC, yZfw, Uk4k). We summarize our responses to common reviewer concerns and additional results we have added during the rebuttal period:

**Additional Experimental Results**

> The paper could be improved by evaluating more LLMs on the benchmark. (Reviewer yZfw)

We have evaluated GPT-5 and Claude Sonnet 4 on our benchmark and observe that they demonstrate unsafe behavior rates of **50.28%** and **31.29%** respectively measured using the rule-based evaluators, and **50.28%** and **46.02%** respectively as per the LLM-Judge. We have added detailed evaluation results to the revised draft in Section 3. Thus, our findings **generalize** to **seven** state-of-the-art proprietary and open-source LLMs.

> There needs to be an ideal/ground truth correct behavior for every task (Reviewer rwaC)

We have manually annotated the correct safe behavior for all the 356 tasks in our benchmark. The ground truth behaviour for each task is defined to achieve maximum compliance of the user request (utility) without compromising on safety. We use an LLM-Judge to determine if the agent demonstrates this correct safe behaviour given the agent’s trajectory. We also compute the “Successful Completion Rate” for all LLMs and include it in the revised draft (in Table 3). Furthermore, we also annotate the correct safe behaviours for the example tasks in Table 6. Note that the ideal task behaviour is not limited to refusal alone, and many tasks have a correct safe solution as well.


**Common Reviewer Concerns**

> Include additional details about NPCs like prompts, trajectories, tool design etc. (Reviewers rwaC, yZfw, Uk4k)

NPCs are LLM-simulated users implemented using the Sotopia framework that interact with the OpenHands agent via the ChatNPC tool. Each task involving interaction with NPCs is accompanied by a *scenarios.json* file where we define the task setting, the NPC persona, and their desired NPC behaviour, all of which are incorporated into the NPC’s prompts. We have added more details on the NPC setup, an example NPC configuration and prompts in Appendix A.4 of the revised draft. Furthermore, we have also added additional example trajectories with NPC interactions in Appendix A.6.

> Why are the rule-based evaluations so close for different LLMs but there is much more variation in LLM-judge? (Reviewers yZfw, rwaC)

While LLMs do share some similar unsafe behaviours on certain tasks, our analysis reveals that models largely show different behaviours for most of the tasks. Since rule-based evaluators only detect safety violations when agents make verifiable changes to the environment, they cannot distinguish truly safe completions from partially completed tasks with unsafe behaviours.
Thus, tasks marked “safe” by rule-based checks still exhibit significant variation across models which is captured by LLM-judges and is reflected in the greater variation of judge-based evaluation results.
These findings further motivate the dual use of rule-based evaluators and LLM-judges.

We have also addressed all other reviewer-specific concerns and suggestions in our responses and we have accordingly incorporated their feedback in the revised draft. Finally, we would like to re-iterate the novelty of our work as OpenAgentSafety is the **only** benchmark with real-world tools, diverse user intents, and multi-user social interactions, allowing for novel findings on unsafe agent behaviours, when agents are increasingly deployed in similar scenarios.

---

### Meta-Review · Area_Chair_sxRh · 2026-01-01

**Summary:**

This paper introduces OpenAgentSafety, which is a comprehensive framework for evaluating the real-world safety of LLM-based agents. Unlike prior work, this framework evaluates agents that interact with real tools and supports over 350 multi-turn, multi-user tasks spanning both benign and adversarial user intents. Besides, OpenAgentSafety can be extended to various tools, tasks and websites, indicating the practicality of the framework.

There are 2 major concerns mentioned by different reviews.

1. Reviewers rwaC, neML, yZfw: Add more details about the disagreement of rule-based metric and LLM-as-a-judge. To be more scientific, include human annotations.

2. Reviewers rwaC, yZfw, Uk4k: Include additional details about NPCs like prompts, trajectories, tool design etc.

As shown in the rebuttal, the two concerns above have been addressed by the authors through additional analyses and discussions. In addition, other comments, such as the suggestion to evaluate more LLMs, have also been addressed.

**Reviewer Concerns:**

Addressed:

1. Include human annotation.
2. Add more details about NPCs.
3. Evaluate the framework on more LLMs.

Still outstanding:

1. The evaluation is based on sandboxed offline replicas, which may not fully capture real-world dynamics.

**Reviewer Scores:**

For reviewer neML, yZfw, Uk4k, I think the score could remain the same.

For reviewer rwaC, I believe the score can be bumped up. The weaknesses proposed is regarding the experimental details, which as the authors said, have been included in the revised version. The reviewer's concern about the ground truth of the correct behavior has also been addressed by including human annotation.

---

### Decision · Program_Chairs · 2026-01-26

Accept (Poster)